# CLEP: Exploiting Edge Partitioning for Graph Contrastive Learning

## Abstract

Generative and contrastive are two fundamental unsupervised approaches to model graph information. The graph generative models extract *intra*-graph information whereas the graph contrastive learning methods focus on *inter*-graph information. Combining these complementary sources of information can potentially enhance the expressiveness of graph representations, which, nevertheless, is underinvestigated by existing methods. In this work, we introduce a probabilistic framework called *contrastive learning with edge partitioning* (CLEP) that integrates generative modeling and graph contrastive learning. CLEP models edge generation by aggregating latent node interactions over multiple overlapping hidden communities. Inspired by the assembling behavior of communities in graph generation, CLEP learns community-specific graph embeddings, which are assembled together to represent the entire graph and further used to predict the graph's identity via a contrastive objective. To relate each embedding to one hidden community, we define a set of community-specific weighted edges for node feature aggregation by partitioning the observed edges according to the latent node interactions associated with the corresponding hidden community. With these unique designs, CLEP is able to model the statistical dependency among hidden communities, graph structures, as well as the identity of each graph; it can also be trained end-to-end via variational inference. We evaluate CLEP on real-world benchmarks under self-supervised and semi-supervised settings and achieve promising results, which demonstrate the effectiveness of our method. Various exploratory studies are also conducted to highlight the characteristics of the inferred hidden communities and the potential benefits they bring to representation learning.

## 1 Introduction

Generative modeling and contrastive learning are both commonly employed to learn graph representations without label supervision. Both types of methods learn the embedding space by leveraging some ground-truth information from the observed graphs, but the aspects of data that each type chooses to fit are different. Graph generative models (Kipf & Welling, 2016; Mehta et al., 2019; Wang et al., 2020) prioritize *intra*-graph information, *i.e.*, the information in each individual graph. The representations provided by graph generative models are usually related to the formation of its own edges. Contrariwise, graph contrastive learning methods (You et al., 2020; 2021; 2022; Xie et al., 2022) focus on capturing *inter*-graph information, they put graphs under comparison to highlight the inherent similarity and differences among a group of graphs.

The difference in the focused graph information leads to complementary strengths and weaknesses of graph generative and contratsive learning methods. The advantage of graph generative models is their ability to recover the structural information of some latent factors, which is lost during graph generation. These latent factors, relevant to each graph in the sense of its own formation, usually preserve valuable information for various graph-analytic tasks. However, the quality of the embeddings provided by graph generative models is questionable because the encoded information is limited to the"expression levels" of these latent factors, which may be insufficient to downstream tasks other than graph generation. Unlike generative models, graph contrastive learning methods cannot automatically find meaningful latent factors in the graph, but they are well recognized for producing high-quality feature representations once the raw structural information is given. An integration of graph generative modeling and graph contrastive learning potentially combines the

complementary strengths of these two classes of methods, which would further benefit representation learning. However, such a direction is insufficiently explored.

In this work, we introduce **C**ontrastive **L**earning with **E**dge **P**artitioning (CLEP), a probabilistic framework that provides a concrete solution to the integration of graph generative modeling with graph contrastive learning. To better leverage the strengths of both methods, we assign different duties to the graph generative model and the contrastive learner, where the former takes the responsibility of extracting the hidden structures of the latent factors which explain graph generation, while the latter treats the inferred hidden structures as new raw inputs, learns their corresponding graph embeddings in a contrastive paradigm, and then aggregates all the embeddings in a way that is intuitively compatible with how the hidden structures interact during graph generation.

More specifically, CLEP is developed upon the graph generative models that explain the formation of edges by cumulative latent node interactions associated with some hidden communities (Yang & Leskovec, 2012; 2013; 2014; Zhou, 2015). Membership and intra-community node interactions are assumed to be independent among different hidden communities. According to the generative model, we embody the structures of these communities by explicitly modeling the average node interactions in each community and partitioning the observed edges accordingly. As indicated by Jenatton et al. (2012), node interactions that happen in different communities may follow multiple relations. It is highly likely that, when nodes interact under different relations, the information exchange that comes along also focuses on different aspects. To better capture the potentially heterogeneous community-specific information, we define a set of encoders to process the information that comes from different communities. The training of these encoders are based on aggregating their corresponding contrastive learning tasks with a set of weights that measure the "importance" of each community. Finally, we gather the community-specific graph embeddings to represent the overall information on a graph, as an analogy with the assembling behavior of communities in graph generation.

We summarize the major contributions of this work as follows:

- We propose CLEP as an integration of graph generative model and graph contrastive learning, which can effectively capture both *intra-* and *inter-*graph information.
- We formulate the statistical problem of training CLEP as the maximum likelihood estimation of a latent variable model, which supports end-to-end optimization via variational inference.
- We show through exploratory studies that the strength of factor-wise representation learning is to capture nonrepetitive graph information from different hidden factors, which offers more flexible embedding selection & combination when facing various downstream tasks.
- We train CLEP under self-supervised and semi-supervised settings and conduct an extensive empirical evaluation of the obtained graph representations, finding that CLEP consistently outperforms existing arts on various real-world benchmarks.

## 2 PRELIMINARIES

**Information encoding on graphs.** A graph encoder maps the information on each graph to a vector representation. For a graph $\mathcal{G}$ with $N$ nodes, its given information usually includes a node feature matrix $\mathbf{X} \in \mathbb{R}^{N \times F}$ and an adjacency matrix $\mathbf{A}$. The most effective way to aggregate these two forms of information is based on graph neural networks (GNNs) (Kipf & Welling, 2017; Hamilton et al., 2017; Veličković et al., 2018; Xu et al., 2019). For a GNN with $T$ layers, denoting $\mathbf{H}_0 = \mathbf{X}$, its propagation rule can be summarized as

$$\mathbf{H}_t = \mathrm{AGG}_t(f_t(\mathbf{H}_{t-1}), \mathbf{A}), \ \ t \in [1, T],$$

where AGG denotes neighborhood aggregation and $f(\cdot)$ denotes nonlinear transformation. Appending a readout layer to a regular GNN converts it from a node-level encoder to a graph-level encoder, where the readout operation is defined as $\mathbf{h} = \mathrm{READOUT}(\{\mathbf{H}_t\}_{t=1,T})$. In the sequel, we use $h_{\mathcal{V}}(\mathbf{A}, \mathbf{X})$ to denote node-level encoders, and use $h_{\mathcal{G}}(\mathbf{A}, \mathbf{X})$ to denote graph-level encoders.

**Graph contrastive learning (GCL).** The ground-truth information that GCL methods use to train the graph encoders is the uniqueness of each graph. The distribution of the graph representations in the embedding space is expected to capture the inherent similarity and differences among the graphs. To this end, positive pairs are created, with each one consisting of two nonidentical views of the same graph. Some contrastive methods (Xie et al., 2022) only pull together the representations of positive

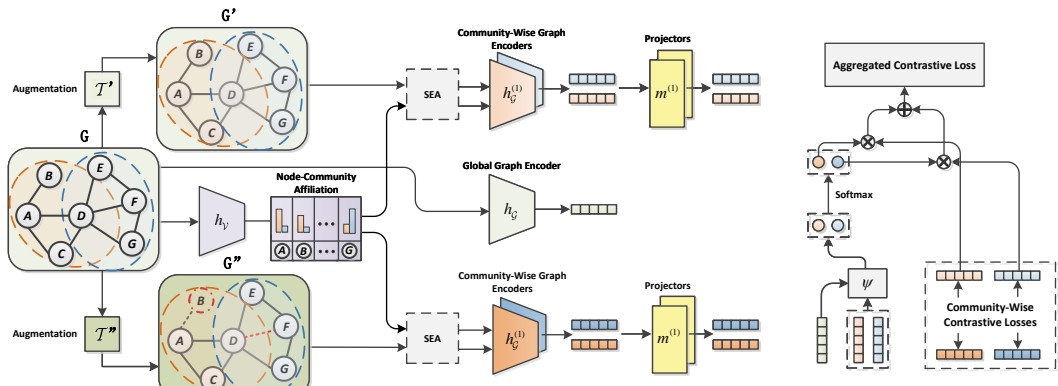

Figure 1: The model architecture of CLEP (left) and the computation of its objective function (right). Node-community affiliation of graph $\mathcal{G}$ is modeled an amortized function. This information is combined with the two augmented views $\mathcal{G}'$ and $\mathcal{G}''$ to partition these graphs in order to reveal the hidden structure of each hidden community in the augmented graphs. The graph is represented by a collection of community-specific embeddings, which are learned by individual contrastive learning tasks customized for the corresponding communities. During training, these tasks are balanced by a set of weights measuring the "relatvie importance" of each community.

pairs, some other methods (You et al., 2020; 2021; 2022) also create negative pairs and push afar the representations of these negative-paired graphs. Unlike the graphs forming positive pairs that share the same origin, graphs in a negative pair are transformed from different source graphs. The update of the embedding space is achieved by optimizing a contrastive objective.

**A probabilistic interpretation.** Many contrastive objectives involving both positive and negative pairs, such as $N$-pair (Sohn, 2016), InfoNCE (Oord et al., 2018) and NT-Xent (Chen et al., 2020), are collectively related to an in-batch instance discrimination event (Wu et al., 2018). Such statistical property is inherited by GCL methods following these established work. Specifically, given a mini-batch $\mathbb{B} = \{\mathcal{G}_1, \mathcal{G}_2, \cdots, \mathcal{G}_{|\mathbb{B}|}\}$, let $\mathbf{k}_i$ be the identity-preserving key of $\mathcal{G}_i$, and $\mathbf{q}$ be a query vector, the probability that the in-batch graph ID of $\mathbf{q}$ being recognized as $i$ can be defined as

$$p(i \mid \mathbf{q}) = \frac{\exp(\mathrm{sim}(\mathbf{q}, \mathbf{k}_i)/\tau)}{\sum_{j=1}^{|\mathbb{B}|} \exp(\mathrm{sim}(\mathbf{q}, \mathbf{k}_j)/\tau)},$$

where $\mathrm{sim}(\boldsymbol{a}, \boldsymbol{b}) := \frac{\boldsymbol{a}^\top \boldsymbol{b}}{\|\boldsymbol{a}\| \cdot \|\boldsymbol{b}\|}$ measures the cosine similarity between vectors $\boldsymbol{a}$ and $\boldsymbol{b}$, and $\tau$ is a temperature parameter. Back to the context of GCL, after each graph $\mathcal{G}$ being augmented into two views $\mathcal{G}'$ and $\mathcal{G}''$, representations $\mathbf{h}'$ and $\mathbf{h}''$ are obtained and mapped to $\mathbf{f}'$ and $\mathbf{f}''$ by a shared parametric function $m(\cdot)$. Treating $\{\mathbf{f}''_i\}_{i=1,|\mathcal{B}|}$ as the keys and $\{\mathbf{f}'_i\}_{i=1,|\mathcal{B}|}$ as the queries (or the other way around) leads to a similar expression to the objective functions in various GCL methods (You et al., 2020; 2021), hence optimizing those objectives can be interpreted as maximizing the log-likelihood of correctly recovering the in-batch graph IDs of the queries.

## 3 CLEP: Contrastive Learning with Edge Partitioning

### 3.1 Latent node interaction and edge generation

Given a graph $\mathcal{G}$ with $N$ nodes, suppose it contains $K$ conceptual latent factors, interpreted as communities $C_1, C_2, \cdots, C_K$ whose intra-community node interactions are recorded as $\{\tilde{\mathbf{M}}^{(k)} \mid \tilde{\mathbf{M}}^{(k)} \in \mathbb{R}_+^{N \times N}\}_{k=1,K}$, the adjacency matrix of $\mathcal{G}$, i.e., $\mathbf{A} \in \{0, 1\}^{N \times N}$, can thus be modeled under the Bernoulli-Poisson link (Zhou, 2015) as

$$\mathrm{A}_{uv} = \vee_{k=1}^K \mathrm{B}_{uv}^{(k)}, \ \mathrm{B}_{uv}^{(k)} = \mathbf{1}(\mathrm{M}_{uv}^{(k)} > 0), \ \mathrm{M}_{uv}^{(k)} \sim \mathrm{Poisson}(\tilde{\mathrm{M}}_{uv}^{(k)}), \ u, v \in [1, N], \ u \neq v. \quad (1)$$

Here $\vee$ denote logical OR. Equation (1) has the following interpretation: for any nodes $u, v$ in graph $\mathcal{G}$, they interact with each other for $\mathrm{M}_{u,v}^{(k)}$ times within community $C_k$, which follows a Poisson

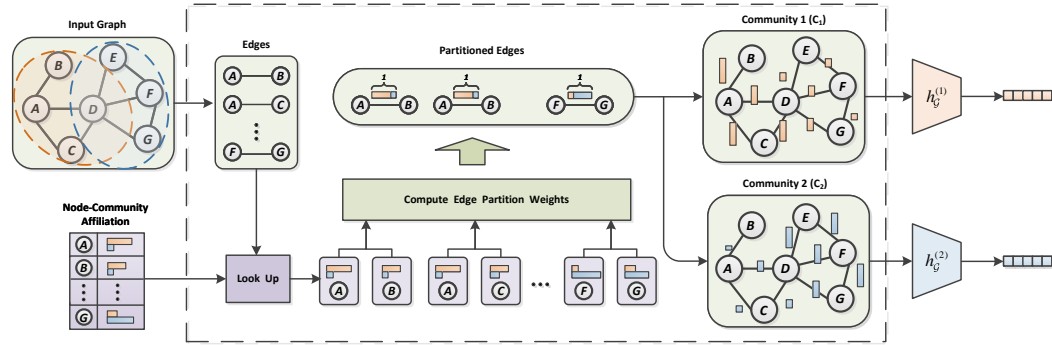

Figure 2: The dashed block encloses the internal steps of soft edge assignment (SEA), which is edge partitioning per se. For each edge, we search for the node-community affiliation of its endpoints and compute the latent interactions of these two nodes in all $K$ hidden communities, then use them to compute the partitioned weights. We embody the hidden structures of the communities with the partitioned graphs, and use them as the raw inputs to learn community-specific graph embeddings.

distribution with rate $\tilde{M}_{uv}^{(k)}$; a latent connection in $C_k$ will be established between $u$ and $v$ if they interact for at least once, and the edge $(u, v)$ can be observed if the latent connection is established in at least one community. Assuming that the communities are mutually independent, we further derive

$$A_{uv} = \mathbb{1}(M_{uv} > 0), \ M_{uv} \sim \text{Poisson}(\tilde{M}_{uv}^{(1)} + \tilde{M}_{uv}^{(2)} + \cdots + \tilde{M}_{uv}^{(K)}), \ u, v \in [1, N], \ u \neq v, \quad (2)$$

as an equivalence to Equation (1). The reformulation explains the formation of edges as a result of accumulating latent node interactions over multiple hidden communities. In practice, to avoid densely parameterizing the edge generation process, each of $\{\tilde{\mathbf{M}}^{(k)}\}_{k=1,K}$ is modeled by a rank-$d$ nonnegative matrix factorization (Yang & Leskovec, 2013; Zhou, 2015), *i.e.*, $\tilde{\mathbf{M}}^{(k)} = \mathbf{Z}^{(k)} \cdot \mathbf{Z}^{(k)\intercal}$, $\mathbf{Z}^{(k)} \in \mathbb{R}_+^{N \times d}$. If $d = 1$, $\mathbf{Z}^{(k)}$ can be interpreted as the affiliation strengths of all nodes to community $C_k$; if $d > 1$, $Z_u^{(k)}$ would be the affiliation strengths of node $u$ to the $d$ sub-communities that are belonged to metacommunity $C_k$. $d$ is by default set to one except being specified otherwise.

### 3.2 Edge-partition-induced community-wise information encoding

In general, to analyze graph information with regard to one type of relation in a multi-relational graph, a critical step is to isolate the edges indicating such relation from the rest of edges (Schlichtkrull et al., 2018; Vashishth et al., 2020). Likewise, when capturing the information on a specific community, we focus on identifying the pairs of nodes which are connected owing to their intense interactions in this community. In CLEP, such goal is achieved by soft edge assignment (denoted by SEA), *i.e.*,

$$A_{uv}^{(k)} = \text{SEA}(A_{uv}, k, \{\tilde{M}_{uv}^{(k)}\}_{k=1,K}) := A_{uv} \cdot \frac{\exp(\tilde{M}_{uv}^{(k)}/\tau_{\text{SEA}})}{\sum_{k'=1}^{K} \exp(\tilde{M}_{uv}^{(k')}/\tau_{\text{SEA}})}, \ k \in [1, K]. \quad (3)$$

Applying Equation (3) to all pairs of nodes yields weighted adjacency matrices $\{\mathbf{A}^{(k)}\}_{k=1}^K$, with $\mathbf{A}^{(k)}$ corresponding to hidden community $C_k$. $\tau_{\text{SEA}}$ is a positive temperature (Hinton et al., 2015) that controls the concentration of the partitioned weights. Setting $\tau_{\text{SEA}}$ close to zero would drive $\{\mathbf{A}^{(k)}\}_{k=1}^K$ towards binary matrices with little overlap on edges, which would highlight the structural difference among the hidden communities. Intuitively, it would encourage extracting the information on the aspects that are unique to each community. Contrariwise, assigning a large value to $\tau_{\text{SEA}}$ would suppress the expression on the differences between the communities.

With the crude relational structural information for all hidden communities embodied by $\{\mathbf{A}^{(k)}\}_{k=1,K}$, we can further refine it by an array of GNN-based graph encoders and obtaining community-specific graph embeddings, *i.e.*, $\mathbf{h}^{(k)} = h_{\mathcal{G}}^{(k)}(\mathbf{A}^{(k)}, \mathbf{X}), k \in [1, K]$. We use the collection of all the community-specific embeddings, *i.e.*, $[\mathbf{h}^{(1)}, \mathbf{h}^{(2)}, \cdots, \mathbf{h}^{(K)}]$ as the overall graph representation for downstream tasks, which has a comprehensive coverage on the information from all hidden communities. A graphic illustration on the computation details of soft edge assignment (SEA) and community-wise information encoding is given in Figure 2.

### 3.3 THE LATENT FACTOR CONTRASTIVE LEARNING MODEL

The statistical problem of fitting CLEP is defined upon mini-batches. Let us denote a generic graph batch as $\mathbb{B} = \{\mathcal{G}_i := (\mathbf{X}_i, \mathbf{A}_i)\}_{i=1,|\mathbb{B}|}$, which is uniformly sampled from the data population. Like many other GCL methods (You et al., 2020; 2021), for any $\mathcal{G} \in \mathbb{B}$, we augment it into two views $\mathcal{G}' = (\mathbf{A}', \mathbf{X}')$ and $\mathcal{G}'' = (\mathbf{A}'', \mathbf{X}'')$ via perturbations $\mathcal{T}'$ and $\mathcal{T}''$. Recalling the probabilistic interpretation of GCL as given in Section 2, the ultimate optimization problem can be defined as

$$\max_{\boldsymbol{\theta}} \mathbb{E}_{\mathbb{B}} \sum_{i=1}^{|\mathbb{B}|} \log p_{\boldsymbol{\theta}}(i \,|\, \mathbf{A}_i, \mathbf{X}_i), \tag{4}$$

where $i$ is the in-batch ID of graph $\mathcal{G}_i$, which serves as the surrogate label of $\mathcal{G}_i$ in the in-batch instance discrimination task. Unlike previous works (You et al., 2020; 2021; 2022) that directly model the probability $p_{\boldsymbol{\theta}}(i \,|\, \mathbf{A}_i, \mathbf{X}_i)$, we take the influence of latent communities into consideration and define the conditional probability $p_{\boldsymbol{\theta}}(i \,|\, \{\mathbf{Z}_i^{(k)}\}_{k=1,K}, \mathbf{A}_i, \mathbf{X}_i)$ instead, *i.e.*,

$$p_{\boldsymbol{\theta}}(i \,|\, \{\mathbf{Z}_i^{(k)}\}_{k=1,K}, \mathbf{A}_i, \mathbf{X}_i) = \sum_{k=1}^{K} p_i^{(k)} \cdot p_{\boldsymbol{\theta}}^{(k)}(i \,|\, \{\mathbf{Z}_i^{(k)}\}_{k=1,K}, \mathbf{A}_i, \mathbf{X}_i), \tag{5}$$

where $p_{\boldsymbol{\theta}}^{(k)}(i \,|\, \{\mathbf{Z}_i^{(k)}\}_{k=1,K}, \mathbf{A}_i, \mathbf{X}_i)$ is the contrastive loss on $\mathcal{G}_i$ that is computed with graph embeddings specified to community $C_k$, and $p_i^{(1)}, p_i^{(2)}, \cdots, p_i^{(K)}$ collectively measure the relative "importance" of each community to graph $\mathcal{G}_i$, subject to the constraint that $\sum_{k=1}^{K} p_i^{(k)} = 1$.

A series of variables are computed in the way as suggested in Figure 1 to facilitate the elaboration on our model. $\mathbf{h}_i = h_{\mathcal{G}}(\mathbf{A}_i, \mathbf{X}_i)$ is a global embedding for the original graph, $\mathcal{G}_i$. The edges of the augmented graphs are soft-assigned to the $K$ communities, *i.e.*, $\mathbf{A}_i'^{(k)} = \mathrm{SEA}(\mathbf{A}_i', k, \{\mathbf{Z}_i^{(k)} \cdot \mathbf{Z}_i^{(k)\mathsf{T}}\}_{k=1,K})$, $\mathbf{A}_i''^{(k)} = \mathrm{SEA}(\mathbf{A}_i'', k, \{\mathbf{Z}_i^{(k)} \cdot \mathbf{Z}_i^{(k)\mathsf{T}}\}_{k=1,K})$; with the partitioned edges, community-specific embeddings of $\mathcal{G}_i'$ and $\mathcal{G}_i''$ can be obtained via $\mathbf{h}_i'^{(k)} = h_{\mathcal{G}}^{(k)}(\mathbf{A}_i'^{(k)}, \mathbf{X}_i')$ and $\mathbf{h}_i''^{(k)} = h_{\mathcal{G}}^{(k)}(\mathbf{A}_i''^{(k)}, \mathbf{X}_i'')$; they are further projected to $\{\mathbf{f}_i'^{(k)}\}_{k=1,K}$ and $\{\mathbf{f}_i''^{(k)}\}_{k=1,K}$ by a group of MLPs, *i.e.*, $\mathbf{f}_i'^{(k)} = m^{(k)}(\mathbf{h}_i'^{(k)})$ and $\mathbf{f}_i''^{(k)} = m^{(k)}(\mathbf{h}_i''^{(k)})$, for $k = 1, 2, \cdots, K$.

With these variables readily available, we specify the quantities in Equation (5) as follows:

$$p_{\boldsymbol{\theta}}^{(k)}(i \,|\, \{\mathbf{Z}_i^{(k)}\}_{k=1,K}, \mathbf{A}_i, \mathbf{X}_i) = \frac{\exp\big(\mathrm{sim}(\mathbf{f}_i'^{(k)}, \mathbf{f}_i''^{(k)})/\tau\big)}{\sum_{j=1}^{|\mathbb{B}|} \exp\big(\mathrm{sim}(\mathbf{f}_i'^{(k)}, \mathbf{f}_j''^{(k)})/\tau\big)}, \tag{6}$$

$$p_i^{(k)} = \frac{\exp\big(\psi(\mathbf{h}_i, \mathbf{h}_i'^{(k)})\big)}{\sum_{k'=1}^{K} \exp\big(\psi(\mathbf{h}_i, \mathbf{h}_i'^{(k')})\big)}, \; k \in [1, K], \tag{7}$$

where $\psi(\cdot, \cdot)$ measures the similarity between the graph embeddings.

### 3.4 THE TRAINING ALGORITHM OF CLEP

We set the prior distributions of the latent variables, namely $P(\mathbf{Z}_i^{(1)}), P(\mathbf{Z}_i^{(2)}), \cdots, P(\mathbf{Z}_i^{(K)})$, as

$$\mathrm{Z}_{i,u}^{(k)} \overset{iid}{\sim} \mathrm{Gamma}(\alpha, \beta), \; k \in [1, K], \; u \in [1, N_i], \; i \in [1, |\mathbb{B}|], \tag{8}$$

where $\alpha, \beta$ are the shape and rate parameters. Although Equations (5) to (7) compute the conditional probability $p_{\boldsymbol{\theta}}(i \,|\, \{\mathbf{Z}_i^{(k)}\}_{k=1,K}, \mathbf{A}_i, \mathbf{X}_i)$, the corresponding density function does not have an analytical expression, hence the true posteriors of $\{\mathbf{Z}_i^{(k)}\}_{k=1,K}$ cannot be explicitly defined, which prohibits direct optimization of the log-likelihood in Equation (4) via Expectation-Maximization (Dempster et al., 1977). Instead, we model $\{\mathbf{Z}^{(k)}\}_{k=1,K}$ by a set of variational distributions $\{Q_{\boldsymbol{\phi}}(\mathbf{Z}_i^{(1)})\}_{k=1,K}$, and maximizing the evidence lower bound, defined as

$$\mathcal{L}_{elbo} = \mathbb{E}_{\mathbb{B}} \left[ \sum_{i=1}^{|\mathbb{B}|} \mathbb{E}_{\mathbf{Z}_i \sim Q_{\boldsymbol{\phi}}(\mathbf{Z}_i)} \log p_{\boldsymbol{\theta}}(i \,|\, \mathbf{Z}_i, \mathbf{A}_i, \mathbf{X}_i) - D_{\mathrm{KL}}(Q_{\boldsymbol{\phi}}(\mathbf{Z}_i) \,\|\, P(\mathbf{Z}_i \,|\, \mathbf{A}_i, \mathbf{X}_i)) \right]. \tag{9}$$

Here we use $\mathbf{Z}_i$ to denote $[\mathbf{Z}_i^{(1)}, \mathbf{Z}_i^{(2)}, \cdots, \mathbf{Z}_i^{(K)}]$ to avoid clutter of notations. The negative KL divergence term can be further expanded as $\mathbb{E}_{\mathbf{Z}_i \sim Q_{\boldsymbol{\phi}}(\mathbf{Z}_i)} \log p(\mathbf{A}_i \,|\, \mathbf{Z}_i) - D_{\mathrm{KL}}(Q_{\boldsymbol{\phi}}(\mathbf{Z}_i) \,\|\, P(\mathbf{Z}_i)) -$

$\log p(\mathbf{A}_i)$. Dropping the constant $\log p(\mathbf{A}_i)$ from Equation (9) yields the final objective of CLEP,

$$\mathcal{L}_{\text{CLEP}} = \mathbb{E}_{\mathbb{B}} \sum\nolimits_{i=1}^{|\mathbb{B}|} \ell_i,$$

$$\ell_i = \mathbb{E}_{\mathbf{Z}_i \sim Q_{\boldsymbol{\phi}}(\mathbf{Z}_i)} \big[ \log p_{\boldsymbol{\theta}}(i \,|\, \mathbf{Z}_i, \mathbf{A}_i, \mathbf{X}_i) + \log p(\mathbf{A}_i \,|\, \mathbf{Z}_i) \big] - D_{\text{KL}}(Q_{\boldsymbol{\phi}}(\mathbf{Z}_i) \,\|\, P(\mathbf{Z}_i)). \tag{10}$$

We define $Q_{\boldsymbol{\phi}}(\mathbf{Z}_i^{(k)})$ as $\text{Weibull}(\mathbf{Z}_i^{(k)} \,|\, \boldsymbol{\Lambda}_i^{(k)}, \mathbf{K}_i^{(k)})$, where $\boldsymbol{\Lambda}_i^{(k)}, \mathbf{K}_i^{(k)}$ are the matrix form of scales and shapes. $\mathbf{Z}_i^{(k)}$ can be sampled via computing the following inverse cumulative density function:

$$Z_{i,u}^{(k)} = \lambda_{i,u}^{(k)}(-\log(1 - \epsilon_{i,u}^{(k)}))^{1/\kappa_{i,u}^{(k)}}, \; \epsilon_u^{(k)} \overset{iid}{\sim} \text{Uniform}(0,1), \; u \in [1, N], \; k \in [1, K]. \tag{11}$$

The shapes and scales are modeled by an amortized function, *i.e.*, $[\boldsymbol{\Lambda}_i, \mathbf{K}_i] = h_{\mathcal{V}}(\mathbf{X}, \mathbf{A})$, where $\boldsymbol{\Lambda}_i := [\boldsymbol{\Lambda}_i^{(1)}, \boldsymbol{\Lambda}_i^{(2)}, \cdots, \boldsymbol{\Lambda}_i^{(K)}]$ and $\mathbf{K}_i := [\mathbf{K}_i^{(1)}, \mathbf{K}_i^{(2)}, \cdots, \mathbf{K}_i^{(K)}]$. With $Q_{\boldsymbol{\phi}}(\mathbf{Z}_i)$ being a Weibull distribution and $P(\mathbf{Z}_i)$ being a Gamma distribution, their KL divergence, *i.e.*, $D_{\text{KL}}(Q_{\boldsymbol{\phi}}(\mathbf{Z}_i) \,\|\, P(\mathbf{Z}_i))$ can be explicitly expressed as in Zhang et al. (2018):

$$D_{\text{KL}}(Q_{\boldsymbol{\phi}}(\mathbf{Z}_i) \,\|\, P(\mathbf{Z}_i)) = \sum\nolimits_{k=1}^{K} \sum\nolimits_{u=1}^{N_i} D_{\text{KL}}(Q_{\boldsymbol{\phi}}(Z_{i,u}^{(k)}) \,\|\, P(Z_{i,u}^{(k)})),$$

$$D_{\text{KL}}(Q_{\boldsymbol{\phi}}(Z_{i,u}^{(k)}) \,\|\, P(Z_{i,u}^{(k)})) = -\alpha \log \lambda_{i,u}^{(k)} + \frac{\gamma\alpha}{\kappa_{i,u}^{(k)}} + \log \kappa_{i,u}^{(k)} + \beta \lambda_{i,u}^{(k)} \Gamma\left(1 + \frac{1}{\kappa_{i,u}^{(k)}}\right) \tag{12}$$

$$- \gamma - 1 - \alpha \ln \beta + \log \Gamma(\alpha),$$

where $\gamma \approx 0.5772$ is the Euler constant. Maximizing Equation (10) with respect to parameters $(\boldsymbol{\theta}, \boldsymbol{\phi})$ would train CLEP. After that, we sample $\{\mathbf{Z}^{(k)}\}_{k=1,K}$ from the well-tuned variational distribution, computing the latent node interactions in each hidden community by $\tilde{\mathbf{M}}^{(k)} = \mathbf{Z}^{(k)} \cdot \mathbf{Z}^{(k)\intercal}$, $k \in [1, K]$, then perform the operations in Section 3.2 to obtain the graph embeddings for downstream tasks.

## 4    OTHER RELATED WORK

**Matrix-factorization-based graph generative models.** A well-established way to build up the stochastic process that generates the adjacency matrix of an observed graph is by low-rank matrix factorization. The seminal idea can be traced back to the stochastic block model (SBM) (Holland et al., 1983), where the latent variables to infer are the assignments of nodes to some latent factors, *i.e.*, communities. SBM models the pairwise node interactions by the dot-product of the node-community assignment matrix and its transpose, which further generate the edges. SBM prohibits nodes to be affiliated with multiple communities, so the latent assignment factors associated with each node are restricted to be one-hot. Such constraint is later relaxed by various extensions of SBM (Airoldi et al., 2008; Miller et al., 2009; Yang & Leskovec, 2012; 2013; 2014; Zhou, 2015; Sun et al., 2019) that permit communities to have overlapped membership. CLEP is closely related to these methods in terms of defining the graph generation process, and the representations obtained by both CLEP and these methods are about the information on these hidden communities. The major difference is that we adopt contrastive learning techniques to embed community-specific information, which brings us immense advantage over these latent factor graph generative models in representation quality.

**Modeling multi-relational data**. Retrieving information from multi-relational data has been extensively studied in the literature of embedding knowledge graphs (Schlichtkrull et al., 2018; Vashishth et al., 2020). Different relation types are expressed by edge labels, which are usually provided as a part of the groundtruth information. A consensus of these models is that different relations should be separately modeled, which is in general carried out by factorizing the graph by edge type, then individually modeling the information on each graph factor that only contain one type of edges. In a similar way, edges in the graph that CLEP deals with are also inherently different with each other in terms of the "major contributor" of their generation, *i.e.*, the community that contributes the most of the interactions, which creates a special case of multi-relational graph. We hence adopt a similar high-level representation pipeline that contains both graph factorization and factor-wise representation learning. However, the necessary information to perform graph factorization is not observable, thus we express it with a set of latent variables and infer them via variational inference.

**Disentangled graph learning.** Isolating the structure of each community from the entire graph then modeling each isolated sub-structure with a customized model relates CLEP to disentangled graph

Table 1: Comparison of graph classification performance, results are reported in the format "average accuracy ± standard error". "A.R." stands for average ranking.

| Method | MUTAG | PTC_MR | PROTEINS | NCI1 | IMDB-B | IMDB-M | RDT-B | RDT-M5K | A.R.↓ |
|---|---|---|---|---|---|---|---|---|---|
| SP | 85.2 ± 2.4 | 58.2 ± 2.4 | 75.1 ± 0.5 | 73.0 ± 0.2 | 55.6 ± 0.2 | 38.0 ± 0.3 | 64.1 ± 0.1 | 39.6 ± 0.2 | 10.4 |
| GK | 81.7 ± 2.1 | 57.3 ± 1.4 | 71.7 ± 0.6 | 62.3 ± 0.3 | 65.9 ± 1.0 | 43.9 ± 0.4 | 77.3 ± 0.2 | 41.0 ± 0.2 | 11.1 |
| WL | 80.7 ± 3.0 | 58.0 ± 0.5 | 72.9 ± 0.6 | 80.0 ± 0.5 | 72.3 ± 3.4 | 47.0 ± 0.5 | 68.8 ± 0.4 | 46.1 ± 0.2 | 9.1 |
| DGK | 87.4 ± 2.7 | 60.1 ± 2.6 | 73.3 ± 0.8 | $\underline{80.3}$ ± 0.5 | 67.0 ± 0.6 | 44.6 ± 0.5 | 78.0 ± 0.4 | 41.3 ± 0.2 | 7.3 |
| MLG | 87.9 ± 1.6 | 63.3 ± 1.5 | $\underline{76.1}$ ± 2.0 | **80.8** ± 1.3 | 66.6 ± 0.3 | 41.2 ± 0.0 | - | - | 4.8 |
| node2vec | 72.6 ± 10.2 | 58.6 ± 8.0 | 57.5 ± 3.6 | 54.9 ± 1.6 | - | - | - | - | 13.3 |
| sub2vec | 61.1 ± 15.8 | 60.0 ± 6.4 | 53.0 ± 5.6 | 52.8 ± 1.5 | 55.3 ± 1.5 | 36.7 ± 0.8 | 71.5 ± 0.4 | 36.7 ± 0.4 | 12.5 |
| graph2vec | 83.2 ± 9.3 | 60.2 ± 6.9 | 73.3 ± 2.1 | 73.2 ± 1.8 | 71.1 ± 0.5 | 50.4 ± 0.9 | 75.8 ± 1.0 | 47.9 ± 0.3 | 8.1 |
| GAE | 84.0 ± 0.6 | - | 74.1 ± 0.5 | 73.3 ± 0.6 | 52.1 ± 0.2 | - | 74.8 ± 0.2 | 37.6 ± 1.6 | 9.7 |
| VGAE | 84.4 ± 0.6 | - | 74.0 ± 0.5 | 73.7 ± 0.3 | 52.1 ± 0.2 | - | 74.8 ± 0.2 | 39.1 ± 1.6 | 9.3 |
| InfoGraph | 89.0 ± 1.1 | 61.7 ± 1.4 | 74.4 ± 0.3 | 76.2 ± 1.1 | 73.0 ± 0.9 | 49.7 ± 0.5 | 82.5 ± 1.4 | 53.5 ± 1.0 | 5.5 |
| MVGRL | 89.7 ± 1.1 | 62.5 ± 1.7 | - | - | $\underline{74.2}$ ± 0.7 | $\underline{51.2}$ ± 0.5 | 84.5 ± 0.6 | - | 3.2 |
| GraphCL | 86.8 ± 1.3 | $\underline{63.6}$ ± 1.8 | 74.4 ± 0.5 | 77.9 ± 0.4 | 71.1 ± 0.4 | 50.7 ± 0.4 | $\underline{89.5}$ ± 0.8 | 56.0 ± 0.3 | 4.4 |
| JOAO | 87.7 ± 0.8 | 61.1 ± 1.7 | 74.6 ± 0.4 | 78.4 ± 0.5 | 70.8 ± 0.3 | 51.0 ± 0.5 | 86.4 ± 1.5 | 56.0 ± 0.3 | 4.5 |
| LaGraph | $\underline{90.2}$ ± 1.1 | - | 75.2 ± 0.4 | 79.9 ± 0.5 | 73.7 ± 0.9 | - | **90.4** ± 0.8 | **56.4** ± 0.4 | 2.2 |
| CLEP | **91.2** ± 0.8 | **65.1** ± 1.2 | **76.4** ± 0.4 | 78.5 ± 0.4 | **75.6** ± 0.4 | **52.0** ± 0.3 | 87.3 ± 0.5 | **56.4** ± 0.3 | 1.8 |

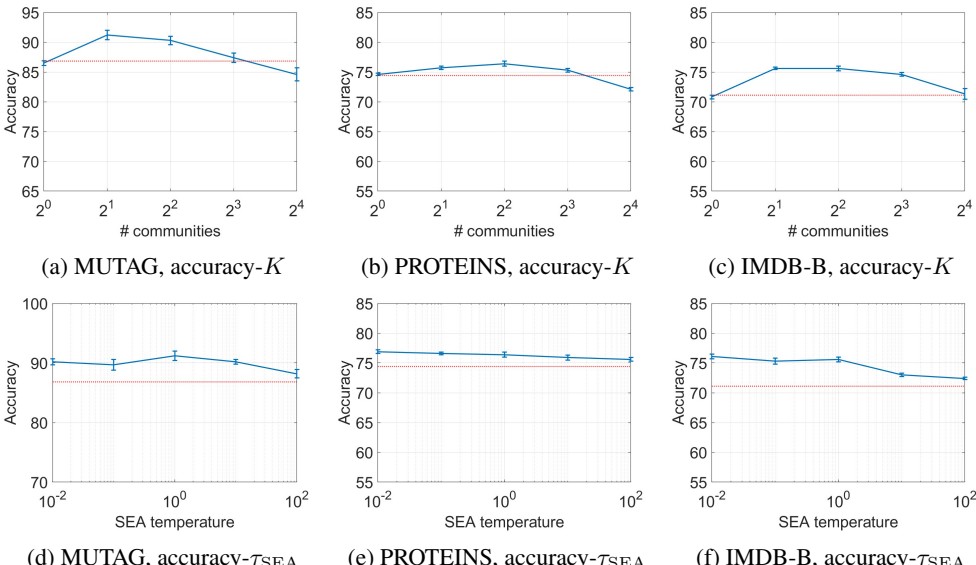

(a) MUTAG, accuracy-$K$  (b) PROTEINS, accuracy-$K$  (c) IMDB-B, accuracy-$K$

(d) MUTAG, accuracy-$\tau_{\text{SEA}}$  (e) PROTEINS, accuracy-$\tau_{\text{SEA}}$  (f) IMDB-B, accuracy-$\tau_{\text{SEA}}$

Figure 3: Task performance with graph embeddings trained with alternative choices of $K$ and $\tau_{\text{SEA}}$, errorbar is set to "± standard error". The red dotted line indicates the performance of GraphCL.

learning (Ma et al., 2019; Yang et al., 2020). In general, the effectiveness of such model design can be explained by better modeling potentially heterogeneous information from different hidden factors, and increased flexibility in terms of the utilization of the information associated with each factor. To make hidden factors more interpretable, this line of work also introduces graph factorization into their frameworks. The major difference between CLEP and disentangle-oriented methods is that the graph factorization in the latter is mostly based on a downstream task, whereas in CLEP, it is based on both graph generation and the pretext contrastive learning task.

## 5 EXPERIMENTS

### 5.1 EMPIRICAL EVALUATION

**Unsupervised learning.** We first examine the quality of the representation learned by only optimizing Equation (10). 8 widely adopted real-world benchmarks are selected for this evaluation task, including 4 biochemical graph datasets (MUTAG, PTC_MR, PROTEINS, and NCI1) and 4

social network datasets (IMDB-BINARY, IMDB-MULTI, REDDIT-BINARY and REDDIT-MULTI). We compare CLEP against three groups of baselines: (i) graph kernel methods, including Shortest Path Kernel (SP) (Borgwardt & Kriegel, 2005), Graphlet Kernel (GK) (Shervashidze et al., 2009), Weisfeiler-Lehman Sub-tree Kernel (WL) (Shervashidze et al., 2011), Deep Graph Kernels (DGK) Yanardag & Vishwanathan (2015), and Multi-Scale Laplacian Kernel (MLG) (Kondor & Pan, 2016); (ii) traditional graph representation learning methods, including node2vec (Grover & Leskovec, 2016), sub2vec (Adhikari et al., 2018), graph2vec (Narayanan et al., 2017), GAE and VGAE (Kipf & Welling, 2016); and (iii) recent state-of-the-arts, including InfoGraph (Sun et al., 2020), MVGRL (Hassani & Khasahmadi, 2020), GraphCL (You et al., 2020), JOAO (You et al., 2021) and LaGraph (Xie et al., 2022). We align most of the hyperparameters with You et al. (2020); for $K$, *i.e.*, the number of communities, and $\tau_{\text{SEA}}$, *i.e.*, the temperature of soft edge assignment, we search $(K, \tau_{\text{SEA}})$ across the grid of $\{2, 4, 8\} \times \{0.1, 1, 10\}$ at the evaluation time through cross-validation. The obtained embeddings are evaluated following You et al. (2020).

We list the results under comparison in Table 1. CLEP has achieved state-of-the-art results on 6 out of 8 benchmarks, and has the best average ranking among all algorithms under comparison. On IMDB-BINARY and IMDB-MULTI, CLEP surpasses the second-best baseline by 1.4% and 0.8%, with both exceeding two standard errors; on MUTAG and PTC_MR, CLEP beats the second place model by 1% and 1.5%, with both exceeding one standard error, indicating a significant improvement to other competitive baselines. CLEP is implemented based on GraphCL, the only difference between CLEP and GraphCL is that the latter does not systematically model the information specified to the hidden communities. When comparing CLEP with GraphCL, the advantage of learning from diverse hidden communities appears to be quite evident, *i.e.*, CLEP outperforms GraphCL by more than one standard deviation on 7 out of 8 benchmarks, at the

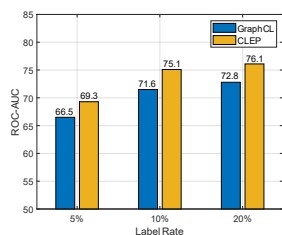

Figure 4: The comparison of classification performance (ROC-AUC) on ogbg-molhiv dataset under the setting of self-supervised learning.

evaluations on MUTAG, PROTEINS, IMDB-BINARY and IMDB-MULTI, the results obtained by CLEP are higher than those of GraphCL by more than two standard errors. These results demonstrate that capturing the information from hidden communities is beneficial for representation learning, and the way that CLEP expresses and encodes the information on each hidden community is effective.

**Semi-supervised learning.** Besides the common setting of unsupervised learning, we compare the developed CLEP with the strong self-supervised baseline, specifically GraphCL (You et al., 2020), under the semi-supervised setting. As shown in Fig. 4, with the learned graph representations on the ogbg-molhiv dataset from Open Graph Benchmark (Hu et al., 2020), we conduct fine-tuning on the partially labeled data, where the settings of label rates are 5%, 10% and 20% respectively, and then evaluate these graph representations on the validation/test sets with binary classification measured by ROC-AUC metric. From the results, we can find that the performance of both methods tends to improve with the increase of the label rate, and our method CLEP achieves significant performance improvement over GraphCL with 2.8%, 3.5%, and 3.3% performance gains by setting the label rate as 5%, 10% and 20% respectively. These results further demonstrate that our CLEP can provide more expressive graph representations even when handling large-scale graph in a semi-supervised manner, benefiting from capturing the information from hidden communities.

## 5.2 ABLATION STUDY

Compared with its base model GraphCL, two hyperparameters, namely $K$, the number of communities, and $\tau_{\text{SEA}}$, the edge partition temperature, are unique to CLEP. The influence of adjusting $K$ or $\tau_{\text{SEA}}$ is shown in Figure 3. In general, within a reasonable range of hyperparameter variation, CLEP outperforms GraphCL most of the time, which shows the consistency of the performance improvement we have achieved against our base model. The optimal values for $K$ appear to be around 2 to 4, and $\tau_{\text{SEA}}$ not greater than 1. When setting $K = 1$, CLEP degenerates to GraphCL, which explains the comparable performances of CLEP and GraphCL; and due to the small size of these graphs, the number of meaningful communities is expected not to be large. As for the selection of $\tau_{\text{SEA}}$, a large value would drive edge partition towards an even partition, leading to a result that the inputs to all community-specific graph encoders are approximately the same as the original graph,

Table 2: Performance on the downstream task, with (w/) and without (w/o) community selection.

|  | MUTAG | PTC_MR | PROTEINS | IMDB-BINARY |
|---|---|---|---|---|
| w/ community selection | **91.4** ± 1.5 | **68.1** ± 1.3 | **75.7** ± 0.4 | **75.4** ± 0.4 |
| w/o community selection | 90.3 ± 1.2 | 67.7 ± 1.4 | 75.4 ± 0.5 | 75.0 ± 0.2 |

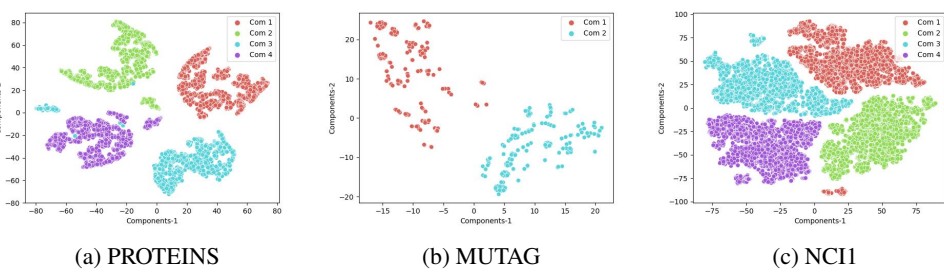

|  (a) PROTEINS  |  (b) MUTAG  |  (c) NCI1  |
|---|---|---|

Figure 5: t-SNE visualization of the community-specific representations learned on some benchmarks.

hence in this scenario, CLEP is analogous to an aggregation of $K$ runs of GraphCL, which cannot effectively improve the quality of the graph embeddings.

### 5.3 EXPLORATORY STUDIES

Shown in Figure 5 are the t-SNE plots of the community-specific graph embeddings obtained from MUTAG, PROTEINS, and NCI1. We color-code the embeddings learned from different communities. From Figure 5 we can observe that points sharing the same color are concentrated at different locations. We interpret the general location of a cluster of colored points as the information "biases" that are related to the nature of the corresponding community category, and the variation across the points in the cluster as the uniqueness of each individual graph. So the spatial clustering of the colored points is a positive sign that CLEP captures community-specific information.

We further investigate how such property of CLEP can be leveraged to benefit downstream tasks. We are interested in the situations where not all communities are needed for the task, *i.e.*, some may be closely related to and can provide crucial information for the task, while the others may be loosely related to the task or even worse, producing more noise than information. The advantage of CLEP is that it organizes the information by communities, which enables the users to actively select & combine the information for different tasks. Intuitively, correctly identifying the "noisy" communities and dropping them from the overall information pool may increase the signal-to-noise ratio of the input information, thus is beneficial for the downstream tasks.

To illustrate the point, we conduct the following experiment on MUTAG, PTC_MR, PROTEINS and IMDB-BINARY. Each dataset is split into 10 folds, with one for validating, one for testing, and the remainder for training. For each benchmark, we fit two logistic classification models, one takes all community-specific embeddings as inputs, and another selects the embeddings by community via cross-validation. The experimental results, as recorded in Table 2, show that community selection results in a consistent improvement to the downstream task. Results in this experiment demonstrate the flexibility of the usage of the graph representations learned by CLEP.

## 6 CONCLUSIONS

In this work, we propose a probabilistic contrastive learning framework called *contrastive learning with edge partitioning* (CLEP). CLEP combines the strengths of graph generative models and graph contrastie learing methods by assigning them different duties that they are adept in, *i.e.*, the graph generative models extracts the hidden communities and thus augments the total amount of information on a graph, and the graph contrastie learning methods are used to convert these hidden structures to high-quality graph embeddings. The empirical evaluation on real-world datasets demonstrate the superiority of the graph representations learned by CLEP. Besides, CLEP organizes the learned graph information by communities, which creates higher flexibility in embedding selection & combining schemes when facing potentially different downstream tasks, which enhances the versatility of CLEP.

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
