# OpenReview forum: "CLEP: Exploiting Edge Partitioning for Graph Contrastive Learning"
_ICLR.cc/2023/Conference — Submitted to ICLR 2023_

### Official Review · Reviewer_zCcf · 2022-10-21

**Confidence:** 5
**Correctness:** 4
**Technical Novelty And Significance:** 4
**Empirical Novelty And Significance:** 3
**Recommendation:** 8

**Clarity, Quality, Novelty And Reproducibility:**

Most of the paper is clearly written, but some parts need more illustration. The paper has good quality in general. The method and the problem studied in this paper are very novel. This paper has a detailed description of its model architecture and training steps, which augment its reproducibility.

**Strength And Weaknesses:**

Strengths:
1. The method proposed in this paper is novel and well-supported, basically, community detection is also an important problem in graph mining, learning the hidden community structure for contrastive learning is an interesting problem.
2. The empirical results are solid, consistent improvements over baselines are demonstrated and the authors perform lots of ablation studies.

Weaknesses:
1. Some parts of the paper are not clearly explained. The authors do not provide an explanation of why Gamma distribution and Weibull distribution are selected in section 3.4. In Table 3, what are task 1 and task 2? Some datasets in Table 3 are themselves binary classification ones, why they are multi-class classification?
2. The graph classification task itself is not that significant, even considering that the number of classes in these datasets is limited. It would be better if the authors could extend the method to other graph mining tasks.
3. The sensitivity analysis on the number of clusters K is missing.

**Summary Of The Paper:**

This paper proposes a new graph contrastive paradigm named CLEP, which learns the latent community structure via variational inference for contrastive learning. It achieves better graph classification performance under both self-supervised and semi-supervised settings. The thorough ablation study also demonstrates the benefits and potential of learning the independent hidden communities.

**Summary Of The Review:**

This paper is very innovative in its method and the problem it studies is also significant. The empirical results are also solid followed by a thorough ablation study. Although some parts need more explanation, this paper in general has a high quality. The main limitation of the model is its applicability, which right now only applies to the graph classification task. I vote to accept this paper, but additional experiments to demonstrate the advantage of the model on other graph mining tasks will be appreciated.

---

> ### Author Response · Authors · 2022-11-18
> **Response to Reviewer zCcf**
>
> Thank you for the commendation of our paper. We are glad that you find learning the hidden structure for contrastive learning an interesting question worth studying. We reply to the weak points you mentioned as follows.
>
> 1. Thank you for pointing out this problem. "task $k$" is intended to express the $k$th one-versus-all task that is separated from a multi-class classification task. For binary classification tasks, the separated one-versus-all tasks actually correspond to the same task. The different experimental results are due to the high variance of the dataset. To reduce the influence of such variance to the intactness of our conclusions, we have updated this experiment as follows: we split each dataset into 10 folds, with train-validation-test ratio as 8-1-1. We record the average of all results obtained with the same initialization, where every fold is treated as the test fold for one time; repeat the same experiments with 10 random initializations. So for each experiment, we obtain 10 results, and we finally report the model performance in the format ``mean of the 10 results $\pm$ standard error of the 10 results''. **We have also updated Section 5.3 accordingly**.
>
> 2. Thank you for the suggestion. We also consider experimenting the CLEP framework on node classification. Since it may lead to new model design, we consider that extension as a separate work by itself.
>
> 3. Thank you for the notice. **We have added the ablation study on $K$ (the number of communities) and $\tau_{\mathrm{SEA}}$ (the temperature parameter for soft edge assignment) in Section 5.2, in the rebuttal revised version**.

---

### Official Review · Reviewer_mFPj · 2022-10-24

**Confidence:** 4
**Correctness:** 3
**Technical Novelty And Significance:** 3
**Empirical Novelty And Significance:** 2
**Recommendation:** 5

**Clarity, Quality, Novelty And Reproducibility:**

This paper is well-written, but please take care of some typos. E.g., what "CEGCL" means in the abstract?

**Strength And Weaknesses:**

Strengths:
1. This paper unifies two unsupervised learning approaches: graph generative models and graph contrastive learning in a probabilistic framework, which is novel to me.
2. The idea to factorize the graph facilitates the representation learning framework, demonstrated by experimental results on downstream classification tasks.

Weaknesses:
1. The motivation to factorize the graph into subgraphs is unclear to me. Since assuming multiple communities will significantly enlarge the number of parameters needed to represent nodes, it is better to justify this complication is worthy.
2. The experimental part is inadequate for me. E.g., (1) More experiments on studying community structures can better justify the proposed method. (2) Since the paper uses a Gamma prior to the latent representation, it should observe some sparsity level within the node representation. Sometimes, sparsified representations can demonstrate a clearer community representation. But a sparsified representation is probably not optimal for classification tasks. It would be better to include discussions on the choice of Gamma hyper-parameters.

**Summary Of The Paper:**

This paper proposes a probabilistic method to combine graph generation and contrastive learning. A key innovation is to factorize the graph into several subgraphs. Each subgraph represents a facet of interactions in a sub-community. The learned latent representation for each node is also factorized into sub-representations w.r.t. each community. Using this fine-grained representation, the paper unifies graph generative models and contrastive learning on the community level, demonstrating better classification performance than previous methods.

**Summary Of The Review:**

This paper proposes a novel idea to combine existing approaches to achieve better classification performance via an augmented representation. The performance gain needs some experimental justification to make it more convincing.

---

> ### Author Response · Authors · 2022-11-18
> **Response to Reviewer mFPj**
>
> Thank you for the thoughtful suggestions and insightful questions. We are glad that community-based graph factorization and representation is recognized as novel. Below we respond to your specific suggestions & questions.
>
> 1. Justification of increased parameters introduced by community-wise representation learning.
>
>    Learning a set of embeddings for each community is analogous to learning $K$ graph embeddings at the same time. Such overhead is created by the goal of community-specific representation learning. By separating the communities and learn different graph embeddings for different communities, we organize the information on a graph by community, so that a downstream task can select or combine such information in a way that is favorable for itself, which
>
>    - leads to better empirical results than our base model GraphCL;
>    - (as shown in Section 5.3) increases the versatility of these embeddings, *i.e.*, users can select embeddings from the communities that are statistically relevant to the task, and the model for downstream tasks can be trained to optimally combine the embedded community-specific graph information.
>
> 2. Adding experiments on the community structure.
>
>    Thank you for the suggestion. We are working on the experiments to reveal the detected communities and the structures we obtained via soft edge assignment, and will add them in future revision.
>
> 3. Rationale of Gamma priors.
>
>    Thank you for the question. The Gamma prior is used to model the strength of node-community affiliation, which intuitively should be a sparse nonnegative quantity and hence can be well fit with Gamma. Although node-community affiliation is treated as node representations in many graph generative models, in CLEP, they are only used to model node interaction rates in each community, which further determines how the edges are soft-assigned. In other words, we do not directly readout node-community affiliation into graph representations, hence the properties of node-community affiliation as node representation are beyond the scope of this work thus not discussed in the paper.
>
>    Another reason is that the Gamma priors allow the use of Weibull variational distributions, which are reparameterizable and have analytic KL divergence when paired with Gamma priors.
>
> 4. Taking care of the typos
>
>    Thank you for pointing out the typos, we have fixed them in revision.

---

### Official Review · Reviewer_AdF9 · 2022-10-26

**Confidence:** 5
**Correctness:** 3
**Technical Novelty And Significance:** 2
**Empirical Novelty And Significance:** 2
**Recommendation:** 3

**Clarity, Quality, Novelty And Reproducibility:**

The clarity is well, including writting and model description.
The quality is relatively high.
The novelty is relatively limited, since two key methods are from existing works.

**Strength And Weaknesses:**

Strength
1. The paper is written well, and the method is described clearly.
2. Combining the idea of graph contaning multiple conceptual community-aware interactions and graph contrastive learning is novel, although these two methods are borrowed from existing works.
3. The probabilistic perspective for the proposed CLEP is clear and robust, including the interpretation of the extended contrastive loss function as well as model optimization via varational inference.

Weakness

Overall, the weakness is primarily about experimental results:
1. Compared with baselines, the improvement in unsupervised classification seems trivial, although the authors claim it's larger than one or two s.t.d. Therefore, the proposed CLEP may not be effective in terms of graph classification.
2. For semi-supervised classification, only comparing with GraphCL is not convincing, and other baselines should be included, like LaGraph, MVGRL, etc.
3. For the ablation study, the influence of K (#communities) should be evaluated, especially for K=1. This is necessary to demonstrate the effectiveness of the proposed community-aware contrastive learning.


**Summary Of The Paper:**

This paper proposes to integrate community-aware graph generation (or edge partition) and graph contrastive learning, which is claimed to encode both intra-graph and inter-graph information. Specifically, the basic contrastive loss is extended based on community-aware edge partition, incorporating contrastive learning on both original-augmented and community-community graph pairs. Extensive experiments are conducted on multiple datasets with unsupervised and semi-supervised graph classification, but the results can not demonstrate the effectiveness of the proposed CLEP.


**Summary Of The Review:**

Although the idea is interesting and the proposed method seems promising, the experiments are not convinced for the effectiveness of model design.

---

> ### Author Response · Authors · 2022-11-18
> **Response to Reviewer AdF9**
>
> Thank you for the thoughtful suggestions on the experiments. **We have added the following ablation studies to the revised version (in Section 5.2)**
>
> 1. an ablation study on $K$, the number of communities;
>
> 2. an ablation study on $\tau_{\mathrm{SEA}}$, the temperature parameter for soft edge assignment.
>
> We are also working on adding other baselines to to the semi-supervised learning experiment, as suggested.

---

### Official Review · Reviewer_9b1o · 2022-10-28

**Confidence:** 4
**Correctness:** 3
**Technical Novelty And Significance:** 2
**Empirical Novelty And Significance:** 2
**Recommendation:** 3

**Clarity, Quality, Novelty And Reproducibility:**

In general, the main idea and the proposed model are clearly explained and technically sound, and a competent researcher would be able to reproduce the model. But the writing of this paper can be improved, since there are many typos, and some important points of the paper are difficult to understand.

**Strength And Weaknesses:**

Strengths:
1. In general, the main idea and the proposed model are clearly explained. The proposed model is described in detail, and a competent researcher would be able to reproduce the model.

2. Intensive experiments have been conducted to verify the performance of the proposed CLEP. Apart from the node classification result on eight real-world benchmarks, this paper also conducted exploratory experiments to highlight the versatility of CLEP on different downstream tasks.

Weaknesses:
1. The motivation of the paper is quite unclear. At the beginning of this paper, the author argues that “An integration of graph generative models and graph contrastive learning methods potentially aggregates both intra- and inter-graph information and combines the complementary strengths of these two classes of models, which would further benefit representation learning”. But what is the definition of intra/inter graph information? And how does such information benefit graph learning? More importantly, how does the proposed framework handle such problems, i.e., is there any tailored component or mechanism proposed to extract intra and inter graph information?

2. The concepts used in this paper are somehow casual and confusing. For example, the authors highlight the “intra- and inter-graph information” in the introduction section but quickly turn around to how to capture “heterogeneous community-specific information” in the whole methodology section. How to define “intra- and inter- graph information” and “heterogeneous community-specific information” in the learning framework? And what is the correlation between these concepts? The continuously introduced confusing concepts make this paper very hard to follow.

3. Some important points of the paper are difficult to understand. For example, the authors summarized the proposed method with the sentence, “To better capture the potentially heterogeneous community-specific information, we define a set of encoders to process the information that comes from different communities” on page 2. Then, first, what’s the community-specific information? Second, why is such information heterogeneous? How to deal with the heterogeneity? Third, how does the proposed CLEP determine the number of communities, i.e., K, for different datasets? Is K a tunable hyperparameter? If so, is the number of communities crucial for the model’s learning capability? If not, is there any tailored mechanism used in this paper to automatically choose the suitable K for different datasets? From the whole paper, I cannot find the answers to the questions listed above.

4. Some important baseline methods are missing. This paper supposes that edges in the graph are built based on K conceptual latent factors. In fact, the same setting has been intensively studied in many disentangled graph learning frameworks, such as DisenGCN(Disentangled Graph Convolutional Networks) and DGCL (Disentangled Contrastive Learning on Graphs).

5. Parameter analysis is totally missing in this paper. As the key hyper-parameter in the proposed model, the positive temperature (\tau_SEA) used in Equation (3) is not well studied.

6. There are many typos. For example, the author mentioned that “Inspired by the “assembly” behavior of communities in graph generation, CEGCL learns community-specific graph embeddings and assemble them together to represent the entire graph” in the abstract. However, it seems that “CEGCL” is a totally new term that has never appeared in this paper.

**Summary Of The Paper:**

This paper proposed a probabilistic graph learning framework named CLEP, which unifies graph generative models with contrastive learning paradigm to extract intra- and inter-graph information. CLEP consists of two major components: 1) the graph generative model, which defines a set of encoders to capture the hidden graph structures from different communities; 2) the contrastive paradigm, which is used to convert the learned hidden structures to high-quality graph embeddings. Intensive experiments have been conducted to verify the performance of the proposed CLEP.

**Summary Of The Review:**

Generally speaking, the core idea of this paper is interesting. However, there are several aspects that raise non-negligible questions about the presented framework.

---

> ### Author Response · Authors · 2022-11-18
> **Response to Reviewer 9b1o**
>
> Thank you for your comments. Below we respond to the weaknesses you have pointed out.
>
> 1. We define *intra-*graph information as the information about a single graph, and *inter-*graph information as the relationship among a group of graphs. The reason that their combination can benefit graph learning is that they focus on different aspects of the given graph dataset, therefore missing any one of the two may limit the ultimate potential of learned embeddings. In CLEP, we use a generative-model-induced soft edge assignment mechanism to partition the observed graph into community graph factors, this part of operations are based on *intra*-graph information; after that, we embed these community graph factor by contrastive learners, since contrastive learning compares different graphs, this part of operations are based on *inter*-graph information. **We have updated the Introduction section** to elaborate these terms.
>
> 2. The high-level idea of CLEP is to find out the hidden structures that give rise to graph generation, embody them with graphs, then embed these hidden structures by means of contrastive learning. The detection and embodiment of these hidden structures utilizes *intra*-graph information, and the embedding step utilizes *inter*-graph information. To implement the high-level idea, in order to design a reasonable model, we need to analyze the characteristics of these hidden structures. In CLEP, we identify these hidden structures as communities, and one characteristic of real-world communities is that they have heterogeneous / different semantic interpretations. For instance, different communities in a social network may correspond to different social groups, and different communities in molecular networks may correspond to different functional groups. Stressing the heterogeneity of information from different communities is a justification of applying different contrastive learners to different communities. **We have updated the Introduction section** to better illustrate this point.
>
> 3. We use the term "community-specific information" to describe the information that we extract from the members and their connections in one community. Given different connection patterns, the information that we extract from different communities is expected to bias towards each community, hence "specified to" the corresponding communities. For instance, if the communities we have separated during soft edge assignment are in reality a sports club and a reading group, we expect to learn information about sports from the sports club community, and information about reading material from the reading group community. The heterogeneity of information extracted from different communities are closely related to the divergent semantic interpretations of these communities. The hyperparameters, namely the number of communities $K$, and the temperature for soft edge assignment $\tau_{\mathrm{SEA}}$, are usually determined via cross-validation in a supervised learning task; however, since CLEP is proposed for self-supervised learning, we do not have the ground-truth labels to conduct cross-validation, so we pre-set several combinations of these two hyperparameters and learn several sets of embeddings. In the testing time, when the validation labels are finally available, we use them to select one set of embeddings. This part of procedures are discussed in the first paragraph of Section 5.1. **We have added ablation studies on these hyperparameter in Section 5.2**. Due to the page limit, we move the original ablation study to the Appendix.
>
> 4. Thank you for providing us with these related models. **We have added DisenGCN as a related work in the rebuttal revised submission**. As for DGCL, although the methodology also sets extracting information about hidden structures in a graph as a goal, we find the technical details given in the paper not sufficient to reproduce the results. We have contacted the authors for the codes but they are not available yet. Hence it is hard to compare CLEP with DGCL at the moment, and we are glad to add DGCL as a baseline once its official implementation is released with verifiable reproducibility.
>
> 5. Please refer to response (3) for details.
>
> 6. Thank you for spotting these typos, we have already fixed them in revision.

---

### Official Review · Reviewer_JYUf · 2022-11-03

**Confidence:** 2
**Correctness:** 2
**Technical Novelty And Significance:** 2
**Empirical Novelty And Significance:** 2
**Recommendation:** 3

**Clarity, Quality, Novelty And Reproducibility:**

Clarity: Fair

Quality: Fair

Novelty: Limited

Reproducibility: Not Sure


**Strength And Weaknesses:**

**Weaknesses**
* I don't understand why the author need to use generative model in contrastive learning (CL) and the ELBO that the author propsed seems unrelated to the CL. Can the author explain that.

* The pipline of the over framework is hard to follow. How to generate the embeddings for downstream tasks?

* The authors claim the community information is important in their case. why is that? How do the author verify that the improvement is comes from adding community information.

* To introduce the community information, this work is similar to clustering base GCL.  Related work [1,2] should be compared.

[1] Prototypical Graph Contrastive Learning

[2] Graph InfoClust: Leveraging cluster-level node information for unsupervised graph representation learning



**Summary Of The Paper:**

In this paper, author introduce a probabilistic framework called contrastive learning with edge partitioning (CLEP) that integrates generative modeling and graph contrastive learning. CLEP models edge generation by cumulative latent node interactions over multiple mutually independent hidden communities.

**Summary Of The Review:**

It quite hard to follow the author's idea, It is seems not significant to introducing the generative model in the context of CL. Overall pipline is not clear.

---

> ### Author Response · Authors · 2022-11-18
> **Response to Reviewer JYUf**
>
> Thank you for reviewing our paper. We hope that our revision and responses can help clarify some misunderstandings and confusions. We sincerely appreciate it if you could re-evaluate our paper.
>
> > why the author need to use generative model in contrastive learning (CL)
>
> **The motivation of introducing generative modeling is given in the second paragraph of the Introduction section**. Generative models and graph contrastive learning leverage different ground-truth information for self-supervised learning, so their combination may enrich the information embedded in the graph representations. Specifically, in CLEP, we use generative models to recover the hidden structures (also referred to as community in our paper) whose information is lost during graph generation, and use graph contrastive learning to embed the information on these hidden structures (communities).
>
> > the ELBO that the author propsed seems unrelated to the CL
>
> The objective function (ELBO) in CLEP is given in Equation (10). **The likelihood component $\mathbb{E}[ \log p_{\theta}(i \, | \, \mathbf{Z}_i, \mathbf{A}_i, \mathbf{X}_i)]$ is in fact an averaged NT-Xent loss** (please refer to Equations (6) -- (7)), *i.e.*, we perform contrastive learning for each hidden community, and aggregate these objectives with a set of weights measuring the importance of each community. The reason that it relates to a statistical classification event and thus can be formalized by a likelihood function is elaborated in Section 2, in the paragraph following "A statistical interpretation".
>
> > How to generate the embeddings for downstream tasks
>
> **The question is answered at the end of Section 3.4**. The steps to obtain graph embeddings are as follows:
>
> - Train CLEP by maximizing Equation (10).
> - Sample $\{\mathbf{Z}^{(k)}\,|\,k \in [1,K]\}$ from the variational distribution. The parameters of the variational distribution are modeled by an amortized function, and sampling from such a distribution can be performed via reparameterization. The graphic illustration of this step is shown in Figure 1, at the purple trapezoid "$h_{\mathcal{V}}$".
> - Compute latent node interaction rates $\tilde{\mathbf{M}}^{(k)} = \mathbf{Z}^{(k)} \cdot \mathbf{Z}^{(k)^\intercal}, k \in [1,K]$.
> - Perform soft edge assignment as in Section 3.2, yielding partitioned graphs $\{\mathbf{A}^{(k)} \, | \, k \in [1,K]\}$.
> - Finally, obtain a set of graph embeddings $\{\mathbf{h}^{(k)} \,|\, \mathbf{h}^{(k)} = h_{\mathcal{G}}^{(k)}(\mathbf{A}^{(k)}, \mathbf{X}), k \in [1,K]\}$.
>
> **We also added two pseudocode algorithm in the Appendix, in the rebuttal revised version**, to better elaborate the steps to learn and obtain graph embeddings with CLEP.
>
>
> > How do the author verify that the improvement is comes from adding community information
>
> Removing community-based graph partition (extracting communities) and encoding (modeling community information) makes our model identical to GraphCL, hence the effect of community information can be shown by the comparison between CLEP and GraphCL. **This point is illustrated in the second paragraph of Section 5.1**.
>
> > The authors claim the community information is important in their case. why is that
>
> Community information is what many latent factor graph generative models, such as SBM and its tremendous variates (including GAE and VGAE) use to represent a graph. Specifically, the embedding produced by these generative models can often be interpreted as the "expression level" of hidden communities in the graph. However, we believe that summarizing the information of a community by a scalar expression level may oversimplify the rich structural information in each community, so after identifying the structures of the communities (via soft edge assignment), we apply a contrastive learner to each community, which is expected to produce better embeddings than the scalar "expression level" for each community.
>
> > To introduce the community information, this work is similar to clustering base GCL. Related work [1,2] should be compared.
>
> Thank you for the recommendation. However, work [1] clusters graphs and samples negative pairs from different graph-level clusters, whereas community is more of a node-level or edge-level concept. The structure that we use to embody a community is "nodes that are affiliated with this community, and edges that are connected because of the interactions taking place in this community". As for work [2], although it indeed involves "node clusters", the work is proposed for learning node embeddings, which is not directly comparable with CLEP that focuses on learning graph representations.

---

### Decision · Program_Chairs · 2023-01-20

**Decision:**

Reject

**Justification For Why Not Higher Score:**

The main concerns reviewers had include marginal/incremental improvements and limited comparisons, and unclear motivation. Overall, post rebuttal reviewers have remained mostly negative.

**Justification For Why Not Lower Score:**

N/A

**Metareview: Summary, Strengths And Weaknesses:**

This paper was reviewed by five reviewers and received 3x3 rating, 1x5 and 1x8. The main concerns reviewers had include marginal/incremental improvements and limited comparisons, and unclear motivation. Overall, post rebuttal reviewers have remained mostly negative.